# Research Advances in Hierarchically Structured PVDF-Based All-Organic Composites for High-Energy Density Capacitors

**DOI:** 10.3390/membranes12030274

**Published:** 2022-02-27

**Authors:** Xiaoyong Zhang, Longyan Zhang, Meng Li, Weixing Chen, Jie Chen, Yan-Jun Liu, Yifei Wang

**Affiliations:** 1Shaanxi Key Laboratory of Optoelectronic Functional Materials and Devices, School of Materials Science and Chemical Engineering, Xi’an Technological University, Xi’an 710032, China; zhangxiaoyong371@163.com (X.Z.); lzqpjzzzzzz@163.com (L.Z.); chenwx@xatu.edu.cn (W.C.); 2School of Science, Lanzhou University of Technology, Lanzhou 730050, China; liuyanerjun@163.com; 3Electrical Insulation Research Center, Institute of Materials Science, University of Connecticut, Storrs, CT 06269, USA

**Keywords:** all-organic composites, PVDF, capacitor, polymer film, breakdown strength, electrical energy storage, charge–discharge efficiency, hierarchically structured design

## Abstract

Polymer film capacitors have been widely applied in many pulsed power fields owing to their fastest energy-released rates. The development of ferroelectric polyvinylidene fluoride (PVDF)-based composites has become one of the hot research directions in the field of high-energy storage capacitors. Recently, hierarchically-structured all-organic composites have been shown to possess excellent comprehensive energy storage performance and great potential for application. In this review, most research advances of hierarchically-structured all-organic composites for the energy storage application are systematically classified and summarized. The regulating strategies of hierarchically structured all-organic composites are highlighted from the perspective of preparation approaches, tailored material choices, layer thicknesses, and interfaces. Systematic comparisons of energy storage abilities are presented, including electric displacement, breakdown strength, energy storage density, and efficiency. Finally, we present the remaining problems of hierarchically structured all-organic composites and provide an outlook for future energy storage applications.

## 1. Introduction

Polymer film capacitors are a kind of capacitor that is wound or stacked with polymer material as dielectric and metal foil or metalized coating as the electrode. The charge and discharge of the dielectric metal foil or metalized coating are based on the process of polarization and depolarization. When the electric field is applied and removed, the energy is stored in the dielectric metal foil or metalized coating in the form of the electric field. Dielectric capacitors are essential passive components in extensive power-pulsed applications owing to their fast charge–discharge capability (μs), long lifespan (10^6^–10^7^ cycles), high power density (W m^−^^2^), light weight, and low cost compared to electrochemical apparatus, as shown in Table 1 [1,2,3]. However, the achieved energy densities of dielectric capacitors are usually lower than those of electrochemical devices such as supercapacitors (>20 J cm^−3^) and batteries (>200 J cm^−3^); for instance, the energy density of commercial nonpolar biaxially oriented polypropylene (BOPP) membrane is merely 4–5 J cm^−3^ on account of its low dielectric constant of 2–3 @1 kHz and electric displacement of ~3 μC cm^−2^ at 600 MV m ^−1^ [4,5,6,7,8,9,10]. The unsatisfactory energy density not only becomes restricts the miniaturization and weight reduction of electronic components, but also raises the manufacturing cost and difficulty [11,12,13].

The large maximum displacement, low remnant displacement, and high breakdown strength are vital to the maximum energy density of dielectrics [14,15]. Polyvinylidene fluoride (PVDF) is polymerized from multiple monomer units (–CH_2_–CF_2_–). The crystalline phase of PVDF consists of α, β, γ, and δ. The β phase exhibits high polarizability. Among the four crystal phase structures, the molecular chain of the β phase structure is in the all-trans conformation (TTTT). The uniformly oriented dipole moments in these crystals, perpendicular to the polymer backbone, can form ferroelectric domains, resulting in the highest polarization intensity of PVDF in the β phase. Exceptionally, the nonlinear ferroelectric polymers, e.g., polyvinylidene fluoride (PVDF) and its co- or terpolymers, own a comparatively greater dielectric constant of 10–50 @1 kHz and maximum displacement (e.g., ~9–10 μC cm^−2^) than those of linear polymers. They are applied as a component layer or polymer matrix to enhance energy density [16,17].

In general, there are three approaches by which the energy storage capability of PVDF-based dielectrics can be promoted, which are single-layered organic–inorganic nanocomposites, topologically structured organic–inorganic nanocomposites, and hierarchically structured all-organic composites (including bi-layered, tri-layered, and multilayered composites) [18,19,20]. The first one mainly focuses on enhancing the maximal displacement via introducing inorganic fillers with high dielectric constants. However, serious damage to the mechanical flexibility, raised remnant displacement, and deteriorative breakdown strength are inevitably induced due to the film processability and inorganic nanofillers’ properties. For hierarchical nanocomposites, the maximum displacement and breakdown strength can be significantly increased by rearranging the distribution of electric fields even at a higher packing concentration. Furthermore, the 0D nanoparticles/1D nanofibers are mainly used as the filler of the polarization layer [20,21,22,23,24], while 2D nanosheets are usually chosen as the filler of the breakdown/insulating layer [25,26,27,28,29]. Although high energy density has been obtained in most single-layer and topologically structured organic–inorganic nanocomposites, their efficiencies (<70 %) at 300–600 MV m^−1^ are obviously inferior to commercial BOPP, which has an ultra-high efficiency of ~98%, resulting from the high conductivity micro-regions and carrier migration paths in the organic–inorganic phase interface regions. Therefore, it is urgent to obtain a processable membrane with favorable integrated performance. Fortunately, the optimized energy density, efficiency, and mechanical reliability performance in hierarchically structured all-organic composites can be concurrently achieved by spatially regulating the organization of the constituents.

Herein, we first classify the advanced polymers and offer a summary of the advances in promoting the comprehensive energy storage capability of hierarchically structured all-organic composites, containing double-layer, triple-layer, and multi-layer composites. The influence of heterogeneous interlayer interfaces and constituent layers on dielectric and energy storage performance is summarized and investigated. Finally, we summarize some existing difficulties with hierarchically structured all-organic composite capacitors for future developments.

## 2. Key Energy-Storage Parameters and Calculation Formula

The energy density (*U*_e_) calculation formula of the dielectrics is:(1)$$U_{e}=\int_{D_{rem}}^{D_{max}}E\;dD$$
where *E* represents the electric field, *D_max_* represents maximum displacement, and *D_rem_* is remnant displacement, as shown in Figure 1. Both a high electrical displacement difference (*D_max_*–*D_rem_*) value and large breakdown strength (*E_b_*) are crucial to obtaining large *U_e_*.

Moreover, the efficiency (*η*) of the dielectrics is:(2)$$\eta =\frac{U_{e}}{U}=\frac{U_{e}}{U_{e}+U_{l}}$$
where *U* is the charged energy density and *U*_l_ is the energy loss, which is the difference value between *U* and *U_e_*.

The *E_b_* is analyzed on the basis of the two-parameter Weibull statistic,
(3)$$P\left(E\right)=1-\exp[-\left(E/E_{b}{)}^{\beta }\right]$$
in which *P*(*E*) represents the probability of electrical failure, *E* represents the determined field, *E_b_* denotes the breakdown strength corresponding to a ≈63% failure probability, and the shape parameter *β* presents data scatter.

In order to achieve the desired comprehensive energy storage performance, researchers have been devoted to developing hierarchically structured all-organic composites composed of two or multiple polymers (i.e., ferroelectric and linear polymers), as shown in the schematic diagram of Figure 2. The optimized *D*_max_–*D*_rem_ value, breakdown strength, energy density, and efficiency can be delivered in hierarchically structured all-organic composites by regulating the layer number, layer thickness, layer order, and compositional ratio.

## 3. Hierarchically Structured All-Organic Membranes

### 3.1. Bi-Layered Structure Membranes

PVDF-based co- or terpolymers are often applied as the matrix on account of their high dielectric constant and energy storage densities [30]. Nevertheless, the high dielectric loss of PVDF-based co- or terpolymers results in undesired efficiencies. Consequently, a two-layer structure can be designed by combining polymers with low dielectric losses to achieve lower ferroelectricity and conductive loss. Chen et al. prepared poly(vinylidene flouoride–trifluoroethylene–chlorofluoroethylene)/polyimide (P(VDF-TrFE-CFE)/PI) bilayer membranes via a stream-casting approach, where the PI with linear polar properties is employed as the bottom layer and P(VDF-TrFE-CFE) with nonlinear ferroelectricity polar properties is employed as the top layer (shown in Figure 3) [31].

The results indicate that the dielectric constant of P(VDF-TrFE-CFE)/PI bi-layered membranes is higher than that of single PI membranes. As expected, the dielectric constant of bi-layered membranes decreases gradually as the PI volume ratio increases. The dielectric loss declines with the rise of PI volume ratios, all of which are located between those of single P(VDF-TrFE-CFE) and PI membranes. The breakdown strength is remarkably promoted by building up bi-layered membranes in comparison to those of single membranes; the synergetic influences of the reallocation of the electric field as well as the interfacial obstruction result in the enhanced breakdown strength and energy storage capabilities for the P(VDF-TrFE-CFE)/PI bi-layered membranes. As a result, both the energy density and efficiency of the bi-layered membranes have been remarkably enhanced in comparison with those of the single P(VDF-TrFE-CFE) membrane. For the bi-layered membranes with an optimized PI volume ratio (50 vol.%), the energy density is 9.6 J cm^−3^, which is 1.32 times greater than that of the single P(VDF-TrFE-CFE) membrane, accompanied by an efficiency of 58%. When the PI volume ratios reach 83 vol.%, the energy density is 11.3 J cm^−3^, which is 5 and 1.5 times than those of the single PI (2.4 J cm^−3^) and P(VDF-TrFE-CFE) (7.3 J cm^−3^) membranes, respectively.

### 3.2. Tri-Layered Structure Membranes

With the intention of further improving energy density as well as efficiency, two polymers can be prepared as sandwich-constructed membranes in comparison to the bi-layered membranes. In sandwich structure, on account of the variance of the dielectric constant and volume conductivity between contiguous dielectric layers, interfacial polarization is able to be built up at the interface of dielectric layers with the purpose of optimizing the maximum electrical displacement. Meanwhile, the introduction of an insulating layer into the multilayer structure can inhibit the charge injection of the electrode and produce a huge potential well for charge carriers. On the other hand, local regions with weak electric fields on the heterogeneous interface can suppress the formation of conductive pathways, effectively improving the breakdown strength of the composite, thus enhancing energy density and efficiency [32,33].

Wang et al. prepared sandwich structure composites with a solution-casting approach; the polymer solutions uniformly coated a clean glass plate layer by layer and were dried at 80 °C for 12 h under a vacuum to obtain sandwich-structured composite membranes, where the PVDF with high breakdown strength is the external layer and the P(VDF-TrFE-CTFE) with a high dielectric constant is the intermediate layer [34], as presented in Figure 4. P(VDF-TrFE-CTFE)/PVDF/P(VDF-TrFE-CTFE) and PVDF/P(VDF-TrFE-CTFE)/PVDF membranes were compared by finite element simulation. The results show that the electric field in P(VDF-TrFE-CTFE)/PVDF/P(VDF-TrFE-CTFE) compounds is mainly concentrated on the PVDF layer, which exceeds the maximum breakdown strength of PVDF. The dielectric constant of the PVDF/P(VDF-TrFE-CTFE)/PVDF membrane is greater than that of a pristine PVDF layer but smaller than that of a pristine P(VDF-TrFE-CTFE) layer. Meanwhile, the PVDF/P(VDF-TrFE-CTFE)/PVDF membrane exhibits greater energy storage abilities than those of pristine PVDF and P(VDF-TrFE-CTFE) membranes with an identical electric field. Consequently, the composite with a concentration of 25 vol.% endows the maximal energy density (20.86 J cm^−3^) at 660 MV m^−1^. The maximal energy density of single PVDF and P(VDF-TrFE-CTFE) membranes are 16.02 J cm^−3^ and 11.57 J cm^−3^, respectively.

With the intention of further improving energy storage density as well as efficiency, linear poly(methyl methacrylate) (PMMA) and ferroelectricity-copolymerized poly(vinylidene fluoride-co-hexafluoropropylene) P(VDF-HFP) were introduced for designing the tri-layered all-polymerized membranes via a simple solution-casting procedure (shown in Figure 5) [35]. Improved energy storage performance is realized by collaborative coalition of intentionally regulating the interfaces and the relative ratios of the constituent layers. The dielectric constant of tri-layered all-polymerized membranes declines monotonously as the PMMA ratio increases, resulting from a lower dielectric constant of PMMA relative to that of P(VDF-HFP). Meanwhile, due to the dielectric dispersion in the P(VDF-HFP), the dielectric loss peak of tri-layered all-polymerized membranes is discovered at approximately 10 MHz, which declines as the PMMA ratio increases. The tri-layered all-polymerized membrane with the optimal PMMA ratio (30 vol.%) possesses a maximum energy density of 20.3 J cm^−3^ and an ultra-high efficiency of 84% compared with those of the blended membranes (17.5 J cm^−3^ & 72%) and the bi-layered membrane (17.5 J cm^−3^ & 77%) determined at 440 MV m^−1^. When the ratio of PMMA is 70%, the efficiency reaches 92%, even at 400 MV m^−1^.

Acrylic rubber dielectric elastomers (DE) are a type of field response electrical activity polymer that goes through shape variation when faced with an electric stimulus. The energy storage characteristics under a low electric field are improved by providing maximum polarization value under the electric field. Chen et al. prepared ternary tri-layered structure membranes, in which a P(VDF-HFP) matrix introduced with a certain amount of DE was used as the external layer and linear PMMA was utilized as the intermediate layer [36], as shown in Figure 6. In the preparation process, the ratio of the core layer to the external layer is fixed at about 70:30 to maintain the high energy efficiency, which was proved formerly. Meanwhile, the DE with different concentrations of 5–50 wt.% was incorporated into the P(VDF-HFP) matrix as the external layer to realize significant enhancement in dielectric displacement and energy density, concurrently. It is well-known that energy loss generally originates in the typical electric displacement-electric field (D–E) loops, including ferroelectricity and conductive loss. Moreover, the conductive loss occupies a significant position in energy loss of the ternary tri-layered structure membranes. Apparently, the ternary tri-layered structure membranes display a comparatively low conduction loss of <1 J cm^−3^ until the DE concentration increases to 30 wt.%. In order to gain high energy density at a low electric field, a high electrical displacement difference value (*D_max_–D_rem_*) is crucial. As a result, the highest *D_max_–D_rem_* value of 8 μC cm^−2^ has been concurrently accomplished in the optimal tri-layered membrane (30 wt.% DE). Higher discharged energy density has been achieved in the ternary tri-layered membranes than in those of the single-layered blends and the constituent polymers. At 300 MV m^−^^1^, the ternary tri-layered membrane exhibits an energy density of 11.8 J cm^−^^3^ in comparison to 10 J cm^−^^3^ for the single-layer blends, 8.3 J cm^−^^3^ for the P(VDF-HFP) matrix, and 5.9 J cm^−^^3^ for the PMMA layer.

PMMA/PVDF blends can be utilized as the external layer of the tri-layered architecture due to their great compatibility and molecular interaction. Conduction loss and energy density are effectively restrained by means of modulating the linear dielectric PMMA loadings, resulting from encumbering the dipole-dipole interactivity and decreasing the charged carrier movement [37,38]. Chen et al. produced tri-layered all-polymerized membranes consisting of PMMA/PVDF blends and DE by using a solution-casting procedure (Figure 7) [39]. The dielectric constant of the resulting membranes declined as the PMMA concentrations increased on account of the much lower dielectric constant of PMMA compared to PVDF. The tri-layered membrane with 70 wt.% PMMA shows a significantly lower dielectric loss of 0.07 versus 0.28 of PVDF at 10 MHz. Moreover, the exceptionality of the layered architectures with numerous interbedded interfaces has been strictly confirmed for realizing the promotion in energy density and enhancing *D_max_–D_rem_* value in comparison to those of membrane blends with the optimal PMMA concentration (30 wt.%). As a result, the tri-layered membrane presents a greatly enhanced energy density of 15 J cm^−3^ and a greater efficiency of 76.5% determined at 350 MV m^−1^ on account of lower conductive loss of 15.49%. In contrast, the conductive loss of membrane blends is 21%, which results in an energy density of 8.2 J cm^−3^ at 300 MV m^−1^ accompanied by an efficiency of 72.6%.

To further improve efficiency, linear PMMA was applied as an external layer to decrease *D_rem_*, and the DE was employed as the intermediate layer to increase *D_max_*. Chen et al. prepared tri-layered membranes of PMMA-DE-PMMA via the solution-casting process (Figure 8) [40]. For the purpose of optimizing *D_max_*–*D_rem_*, energy density, and efficiency concurrently, the thickness of the constituent layer can be managed in the casting process. It is worth noting that the optimized DE component ratio has a certain influence on the *E*_b_ of the synthesized film, and the highest *E*_b_ of the optimized tri-layered membrane (10.7 vol.%) is 347.8 MV m^−1^. Furthermore, the maximum *D_max_–D_rem_* value of the PMMA-10.7% is 7.39 μC cm^−2^, which is 56.9% higher than that of the pristine PMMA (4.71 μC cm^−2^). As expected, the maximal energy density of 12.45 J cm^−3^ was obtained, which is 211% greater than that of BOPP (~4 J cm^−3^), even at 640 MV m^−1^. The efficiency of PMMA-10.7% is higher than 90% at 300 MV m^−1^. Even at a temperature of 80 ℃, both a high energy density of ~10.3 J cm^−3^ and an efficiency of 88% were achieved in the PMMA-10.7%.

### 3.3. Multilayered Hierarchical Membranes

Multilayer membranes are reasonably designed, consisting of two or more polymers stacked together. By virtue of the progressive increase in the interface layer, optimized interfacial polarization and electric field distribution can be achieved in multilayer membranes. The heterogeneous interface can more effectively inhibit conductive path formation and improve breakdown strength to promote the energy density and efficiency of multilayered structure membranes.

Xie et al. prepared multilayered all-polymer membranes with alternating PVDF and P(VDF-TrFE-CTFE), as displayed in Figure 9 [41]. Three-layer, five-layer, and seven-layer structure membranes with P(VDF-TrFE-CTFE) volume fractions of 30, 38, and 42 vol.% were prepared. The multilayered all-organic membranes combined the respective advantages of P(VDF-TrFE-CTFE) with a high dielectric constant and PVDF with high breakdown strength [42]. Adopting PVDF as the external layer enables a decrease of the infusion of charges, and the PVDF/P(VDF-TrFE-CTFE) interfaces can encumber the proliferation of the electrical trees, resulting in the enhanced breakdown strength of the multilayered all-organic membranes [43,44]. It can be concluded that the dielectric constant of the multilayered all-polymer membranes is located between pristine PVDF and P(VDF-TrFE-CTFE). Moreover, the dielectric loss of the multilayered all-polymer membranes slightly raises as the number of layers increases. When the number of layers increases, the *D_max_*–*D_rem_*, breakdown fields and energy density are reduced accordingly. The breakdown strength of the membranes with three, five, and seven layers are 620, 443, and 408 MV m^−^^1^, and the largest polarizations are 12.57, 10.22, and 8.69 μC cm^−2^. Although the three-layer membranes have a relatively low dielectric constant, they obtain the highest maximum polarization due to the highest breakdown strength. As a result, the maximum energy density of 18.12 J cm^−3^ is realized in the three-layered PVDF/P(VDF-TrFE-CTFE) membrane, and efficiency is about 80% at 620 MV m^−1^.

In order to further decrease the loss and enhance the energy storage performance of multilayered all-polymer membranes, PVDF polymer was replaced by linear PMMA. Zhang et al. synthesized alternating multilayers by combining PMMA with P(VDF-HFP) via stacking method (shown in Figure 10) [45]. Experiments show that the dielectric constant of the multilayer membranes is about 2 times that of pristine PMMA. Concurrently, as the number of alternating layers increases, the dielectric constant of the multilayered membranes mildly increases on account of the positive impact of interfacial polarization at high frequencies. Nevertheless, the dielectric loss of the multilayered membranes obviously decreases at high frequencies. Owing to the promotion of the mechanical performance and the reduction in the leakage current of the multilayer membranes, the breakdown strength is remarkably enhanced in comparison to that of P(VDF-HFP). As the number of layers increases, the longitudinal evolution and horizontal extension of the electrical trees are effectively restrained. Ultimately, an ultrahigh efficiency of 84.3% and energy density of 25.3 J cm^−3^ can be obtained from the M-9L film, which is 1.34 times that of P(VDF-HFP) (18.8 J cm^−3^ at 569 MV m^−1^) and 2.25 times that of PMMA (11.2 J cm^−3^ at 616 MV m^−1^). The energy densities of 3L (19.9 J cm^−3^ at 633 MV m^−1^), 5L (22.3 J cm^−3^ at 679 MV m^−1^), and 7L (23.4 J cm^−3^ at 698 MV m^−1^) are also higher than that of single P(VDF-HFP) and PMMA membranes.

Cui et al. obtained a PMMA/PVDF all-polymer membrane with a ferroconcrete-like architecture via innovatively combining coaxial spinning with hot compression. During the procedure of spinning, PMMA is enclosed as a shell layer around the PVDF core layer (Figure 11) [46]. The hot compression temperature can be regulated to make sure that the PMMA shapes a constant phase while the PVDF can maintain the initial fiber form. Firstly, the electric field in the PMMA/PVDF all-polymer membrane is reallocated due to the remarkably dielectric invariable variance between PVDF and PMMA. Secondly, by means of regulating the core−shell ratio in the coaxial architecture, the linear PMMA concentration is improved, while the ferroelectricity loss of PVDF can be decreased. Thirdly, this structure adds linear and ferroelectric micro-interfaces, and these interfaces can act as operative charge traps for the purpose of decreasing conductivity, eventually promoting breakdown strength and energy density. As a result, the PMMA/PVDF all-organic membranes exhibit a higher energy density than single PVDF and PMMA membranes; the energy density of 51% PMMA/PVDF ferroconcrete-like membrane is 20.7 J cm^−3^ and the efficiency is 63% at 630 MV m^−1^. Moreover, 45% PMMA/PVDF membrane exhibits an energy storage density of 17.7 J cm^−3^ and an energy efficiency of 73% at 640 MV m^−1^.

## 4. Summary and Outlook

In conclusion, the energy-storage performance, including breakdown strengths, energy densities, and efficiency in hierarchically all-polymerized membranes for high-energy density capacitors are summarized in Table 2. Although many breakthroughs have been made in the past few years, some existing problems still need to be solved in the future, including improving comprehensive energy storage performance, exploring conductivity/breakdown mechanisms, developing high temperature-resistant polymer, and achieving large-scale fabrication.

Hierarchically structured all-organic composites have higher breakdown strengths than those of single-layered and blended films. The electronic breakdown theory can mainly be described by finite element/phase–field simulation [47,48]. Both thermal and mechanical breakdown theories are still in their infancy. Considering that the cross-section of the prepared films is micron-sized, it is very difficult to obtain avisualization of the internal breakdown process.The obtained energy density of hierarchically structured all-organic composites is still at least an order of magnitude smaller than those of lithium-ion batteries and supercapacitors (~200 J cm^−3^). Therefore, there is still room to further enhance energy storage via optimizing structural design.With the pervasiveness of electric vehicles, photovoltaic power generation, and oil and gas exploration and exploitation, high-temperature polymer capacitors are urgently needed for satisfying the demands of harsh and extreme-temperature environments. For instance, the application temperature of a pulse power supply is 125–180 °C, the environmental temperature of oil and gas exploration is up to 175–250 °C, and the ambient temperature of a power inverter of the hybrid electric vehicle can exceed 140 °C.In general, hot pressing [49,50], solution casting [51,52], and melt co-extrusion [53] technology can be used for preparing hierarchically structured all-organic composites. However, there are still some respective shortcomings of the above methods in preparing high-quality and large-scale films. It is difficult to obtain all-organic composites with a nano-sized layer via hot pressing and solution casting methods [54,55]. There are many difficulties in regulating the thickness ratio of multilayered all-organic composites via the melt co-extrusion method. In addition, the price of fluoropolymers and cost of various preparation methods also need to be considered for large-scale film production at the technical level.

Further advancement in these highly transdisciplinary areas rests crucially on the triumphantly synergetic endeavor of synthetic chemists, material specialists, and electrical experts. It is expected that enhanced primary comprehension, modified analog approaches, and groundbreaking energy preservation performance will be accomplished in reasonably planned hierarchical all-polymer membranes in the future.

## Figures and Tables

**Figure 1 membranes-12-00274-f001:**
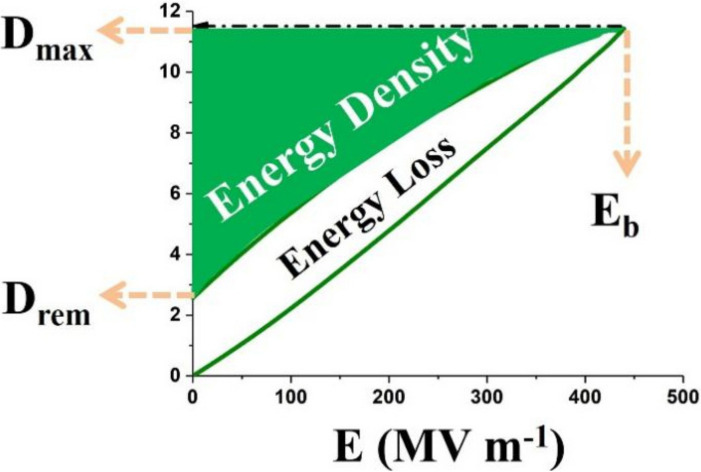
A schematic diagram of electric displacement–electric field (D–E) loops, energy density, and energy loss of PVDF-based dielectrics.

**Figure 2 membranes-12-00274-f002:**
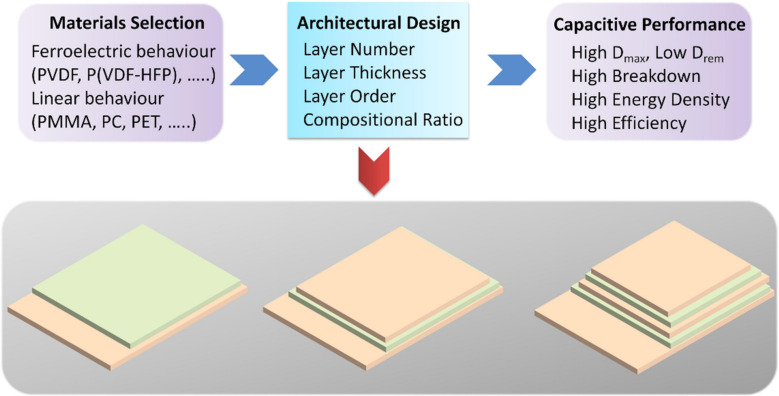
The materials selection, structure design, energy storage performance, and schematic diagram of hierarchically structured all-organic membranes.

**Figure 3 membranes-12-00274-f003:**
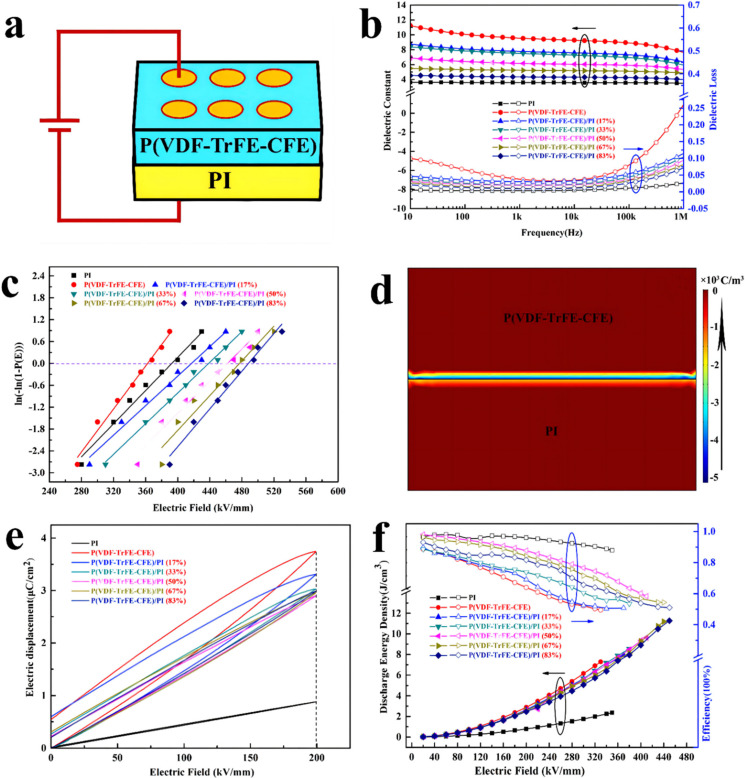
(**a**) Schematic diagram of P(VDF-TrFE-CFE)/PI bi-layered films. (**b**) Dielectric properties of P(VDF-TrFE-CFE)/PI bi-layered films: frequency-dependent changes of the dielectric constant and dielectric loss. (**c**) Weibull breakdown strength of P(VDF-TrFE-CFE)/PI bi-layered films. (**d**) The space charge density simulation for P(VDF-TrFE-CFE)/PI bi-layered films. (**e**) D−E loops of P(VDF-TrFE-CFE)/PI bilayer films with different PI volume ratios at an electric field of 200 MV m^−1^. (**f**) Relationship between the applied electric field and energy storage performances of P(VDF-TrFE-CFE)/PI bi-layered films. Adapted with permission from [31], Copyright © 2022 American Chemical Society.

**Figure 4 membranes-12-00274-f004:**
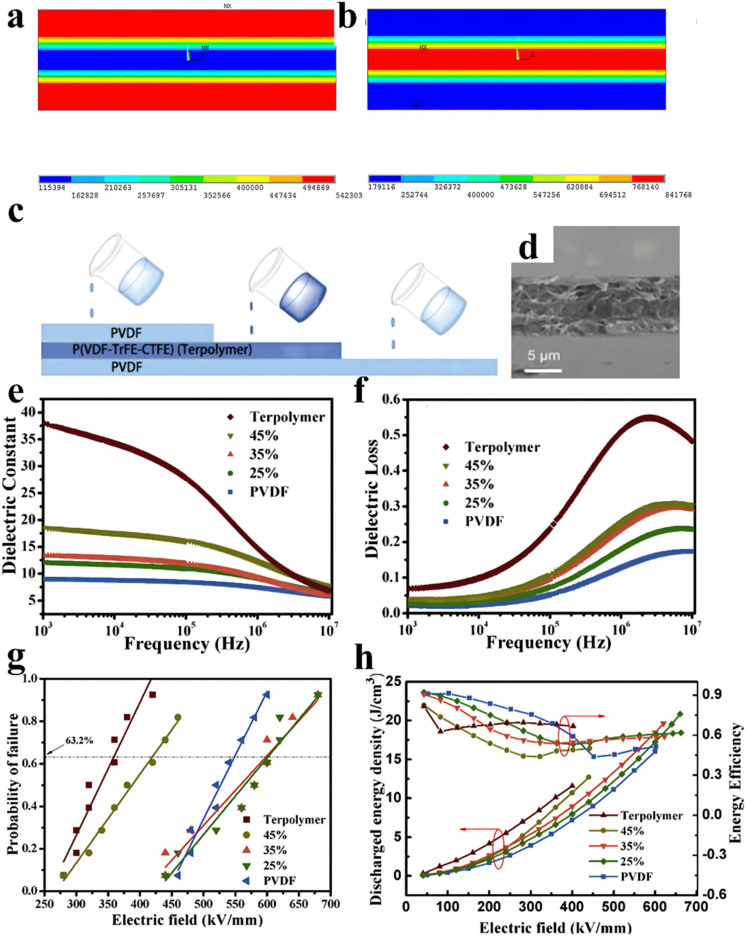
Finite element simulation of the distribution of the electric field in (**a**) the PVDF/P(VDF-TrFE-CTFE)/PVDF composites and (**b**) the P(VDF-TrFE-CTFE)/PVDF/P(VDF-TrFE-CTFE) composites. (**c**) Schematic of the fabrication process. (**d**) Cross-section SEM image of sandwich-structured composites with various contents of P(VDF-TrFE-CTFE): 35 vol.%. (**e**) Dielectric constant. (**f**) Dielectric loss of sandwich-structured composites as a function of frequency. (**g**) Weibull distribution of the breakdown strength. (**h**) Discharged energy density and energy efficiency of pure PVDF, terpolymer, and sandwich-structured composites with different contents of terpolymers. Adapted with permission from [34], Copyright © 2022 Elsevier.

**Figure 5 membranes-12-00274-f005:**
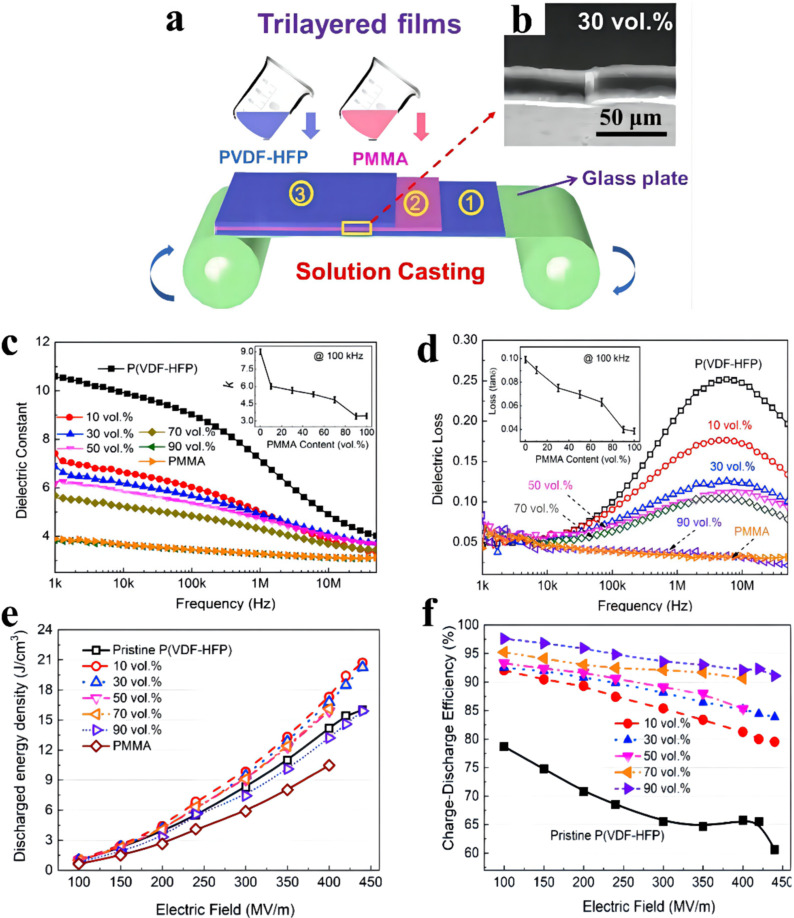
(**a**) Schematic illustration of the tri-layered all-polymer films and (**b**) SEM image of the cross-section of the tri-layered all-polymer film with 30 vol.% PMMA. Frequency-dependent (**c**) dielectric constant and (**d**) dielectric loss of the tri-layered all-polymer films with different PMMA volume fractions. (**e**) Charge–discharge efficiency. (**f**) Discharged energy density. Adapted with permission from [35], Copyright © 2022 Elsevier.

**Figure 6 membranes-12-00274-f006:**
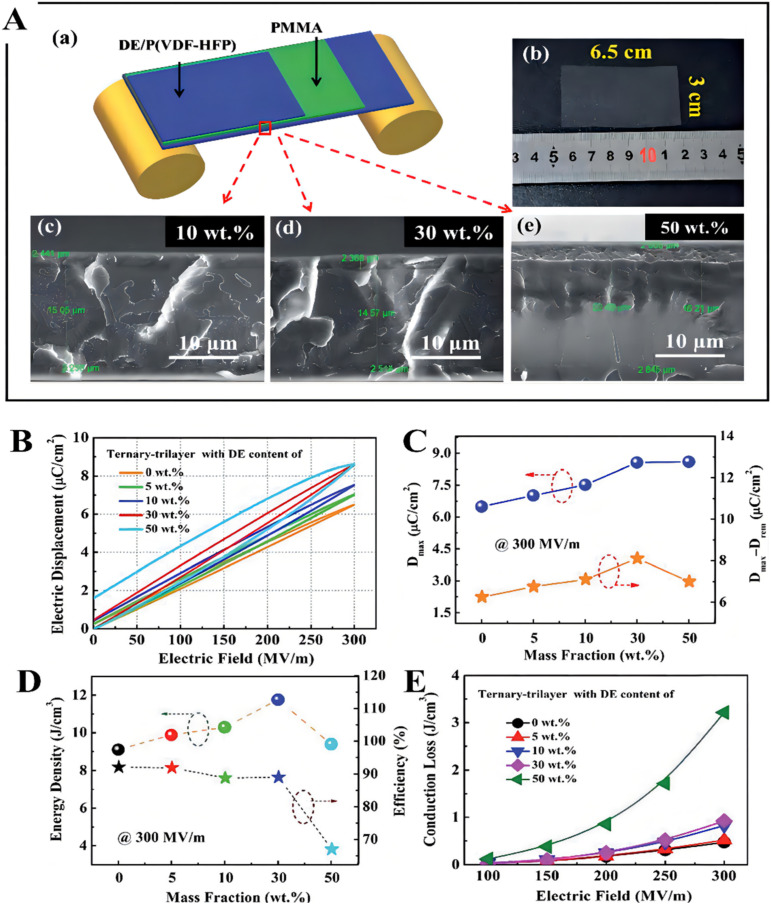
(**A**) (**a**) Schematic illustration for ternary tri-layered architecture films that underwent the layer-by-layer solution casting method. (**b**) Sample photo image of the ternary tri-layered architecture film with 30 wt.% DE and (**c**–**e**) cross-sectional SEM morphologies of ternary tri-layered architecture films with 10 wt.%, 30 wt.%, and 50 wt.% DE films under high magnification, respectively. (**B**) Unipolar electric displacement–electric field (D–E) loops. The typical electric displacement–electric field (D–E) loops of the ternary tri-layered architecture membranes are measured with a 10-Hz unipolar triangle signal as a function of the applied field. (**C**) Maximum and electric displacement difference measured at 300 MV m^−1^ and 10 Hz. (**D**) Discharged energy density and efficiency measured at 300 MV m^−1^. (**E**) Conduction loss at varied electric fields of the ternary tri-layered films at different DE loadings. Adapted with permission from [36], Copyright © 2022 Royal Society of Chemistry.

**Figure 7 membranes-12-00274-f007:**
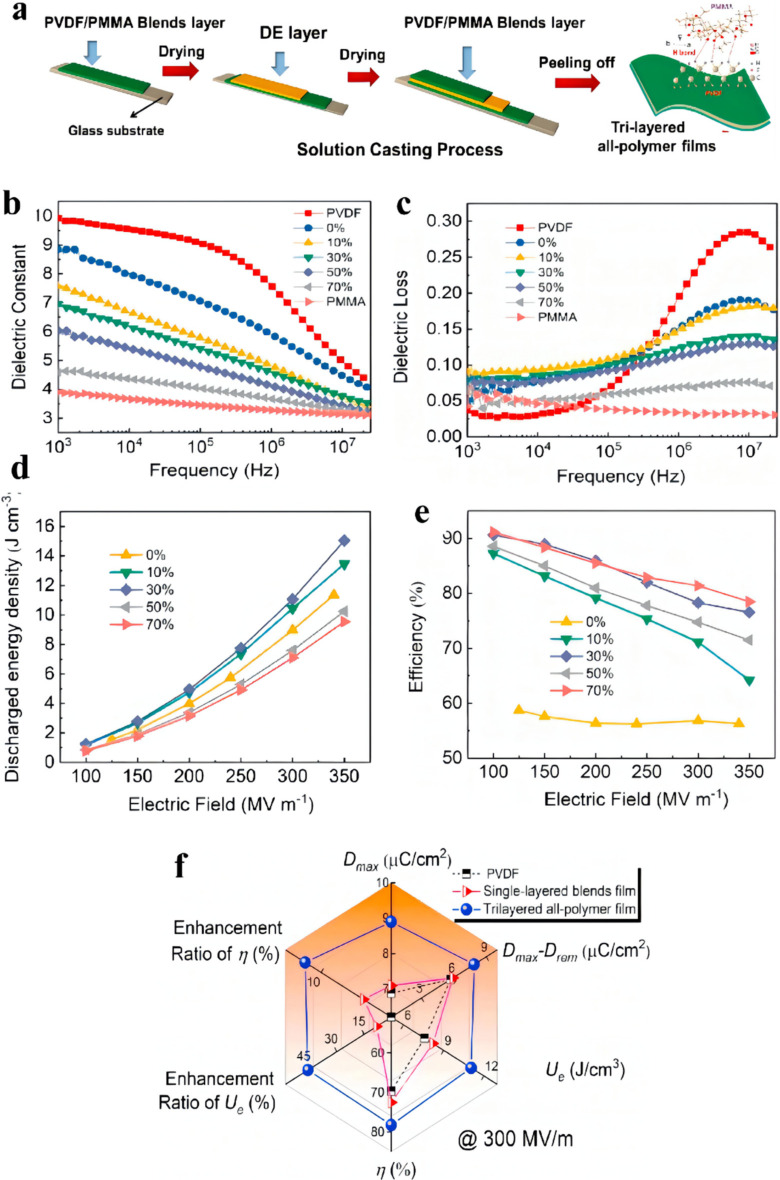
(**a**) Preparation diagram of the solution-processed solution casting process. (**b**) Frequency-dependent dielectric constant. (**c**) Dielectric loss. (**d**) Energy density under varied electric fields. (**e**) Efficiency under varied electric fields. (**f**) Comparison of *D*_max_, *D*_max_–*D*_rem_, *U*_e_, *η*, the enhancement ratio of *U*_e_ and *η* among pristine PVDF, single-layered blends’ film, and tri-layered all-polymer film. Adapted with permission from [39], Copyright © 2022 Elsevier.

**Figure 8 membranes-12-00274-f008:**
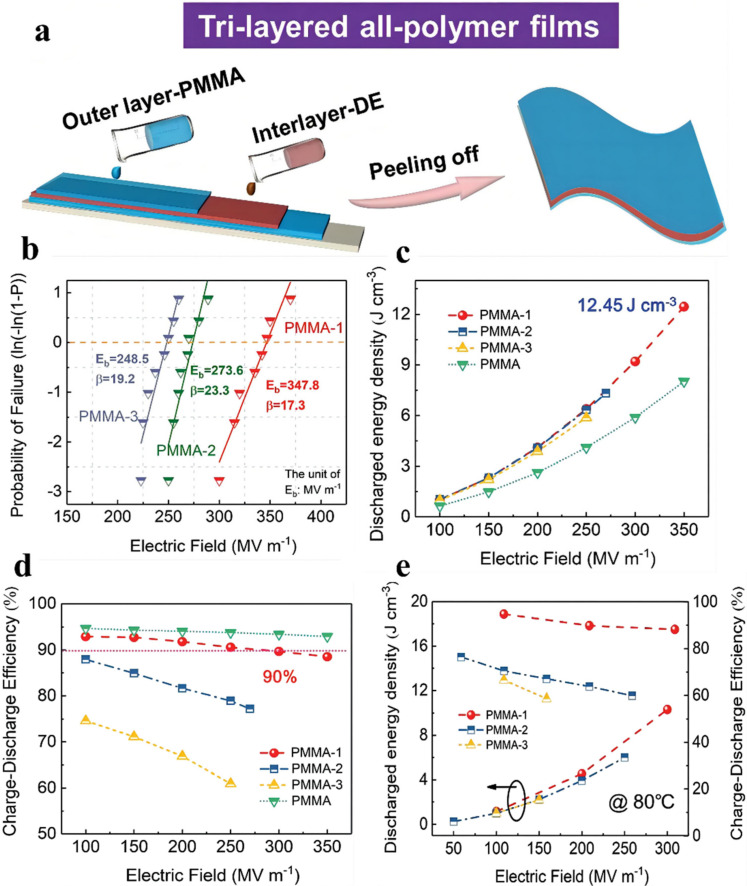
(**a**) Schematic of the preparation process via the layer-by-layer solution casting process. (**b**) Weibull breakdown field distribution. (**c**) Discharged energy density, (**d**) charge-discharge efficiency under varied electric fields of the tri-layered all-polymer composites. (**e**) Comparison of the discharged energy density and charge–discharge efficiency of the tri-layered all-polymer composites at 80 °C. Adapted with permission from [40], Copyright © 2022 Royal Society of Chemistry.

**Figure 9 membranes-12-00274-f009:**
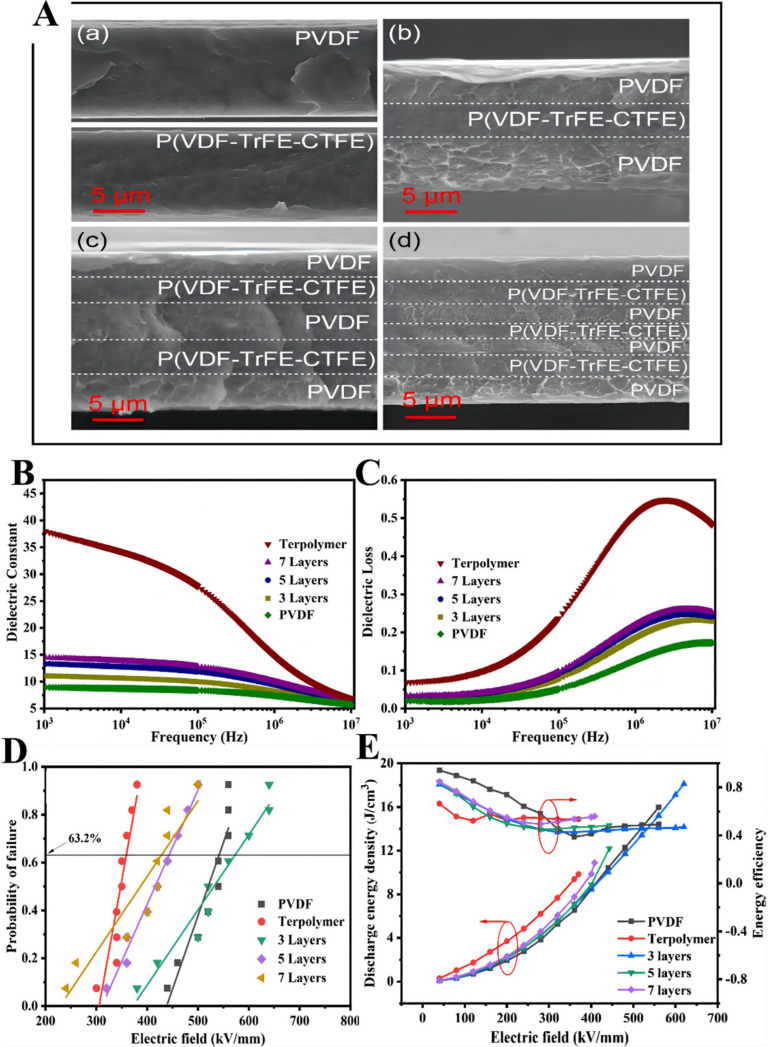
(**A**) SEM images of the cross-sections of (**a**) pure PVDF and P(VDF-TrFE-CTFE), (**b**) three-layer composites, (**c**) five-layer composites, and (**d**) seven-layer composites. (**B**) Dielectric constant and (**C**) dielectric loss of pure polymers and multilayer-structured composites as a function of frequency. (**D**) Weibull distribution of the breakdown strength. (**E**) Energy density and charge–discharge efficiency of pure polymers and multilayer-structured composites. Adapted with permission from [41], Copyright © 2022 American Chemical Society.

**Figure 10 membranes-12-00274-f010:**
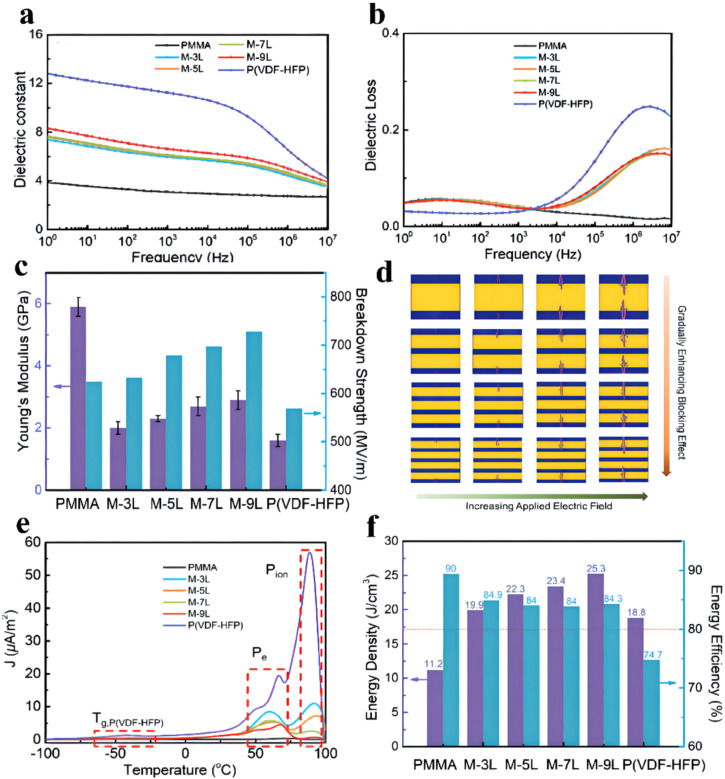
(**a**) Frequency dependence of the dielectric constant and (**b**) dielectric loss for the pure polymer and multilayer composite films. (**c**) Young’s modulus and breakdown strength for the pure polymer and multilayer composite films. (**d**) Breakdown evolution procedures for the multilayer composite films with 3L, 5L, 7L and 9L via phase-field simulation. (**e**) TSDC spectra for the pure polymer and multilayer composite films. (**f**) Comparison of the energy storage properties of the multilayer composite films. Adapted with permission from [45], Copyright © 2022 The Royal Society of Chemistry.

**Figure 11 membranes-12-00274-f011:**
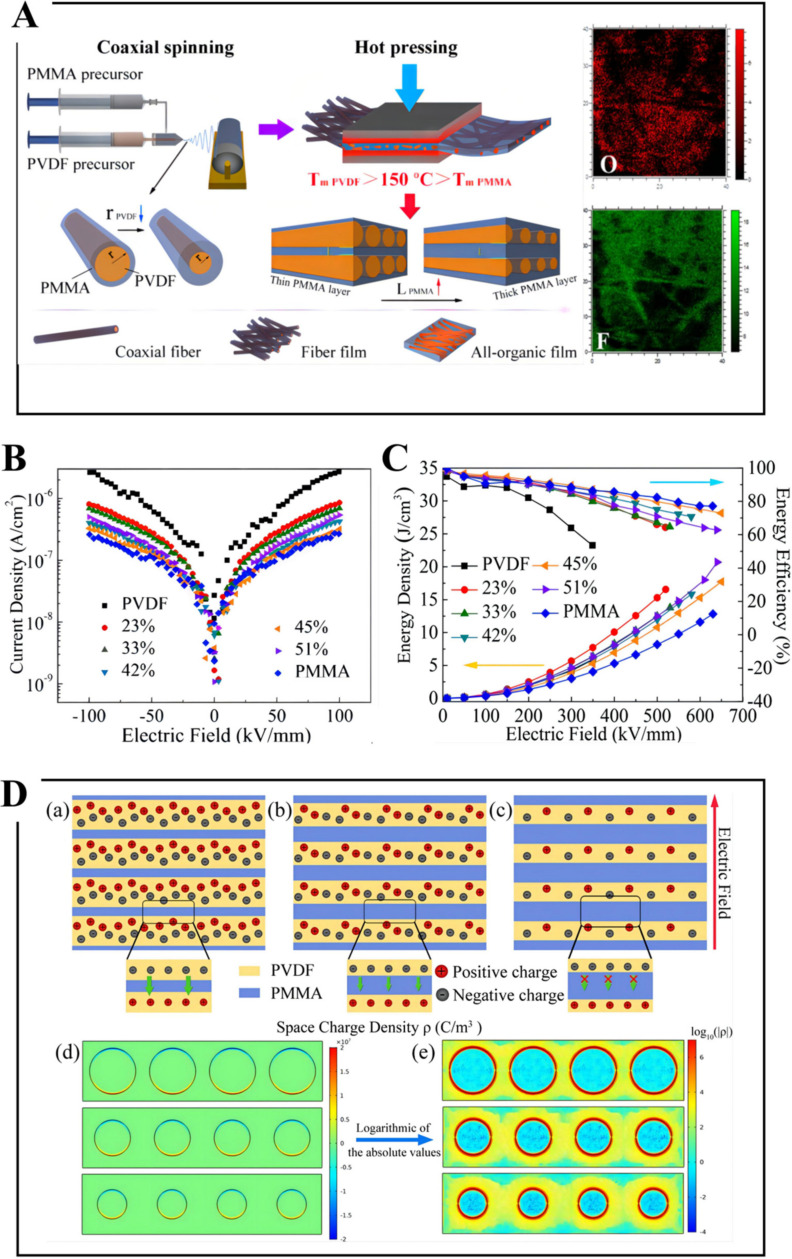
(**A**) Schematic illustration of the fabrication process of PMMA/PVDF all-organic films via coaxial spinning and hot pressing, and time-of-flight secondary ion mass spectrometry (ToF-SIMS) elemental maps of O and F of a 51% PMMA/PVDF all-organic film. (**B**) Current density, (**C**) energy density and energy efficiency of PMMA/PVDF all-organic films with varying PMMA content. (**D**) Schematic of charge migration in local micro-interfaces of a ferroconcrete-like structure. PMMA/PVDF all-organic film featuring (**a**) coarse, (**b**) intermediate, and (**c**) fine PVDF fibers. (**d**) Calculated profile of the space–charge density profile in a localized area of the PMMA/PVDF organic film and (**e**) logarithm of the absolute value of the space–charge density in a local area. Adapted with permission from [46], Copyright © 2022 American Chemical Society.

**Table 1 membranes-12-00274-t001:** Comparison of performance indicators for a variety of energy storage devices.

Energy Storage Devices	Power Density (W kg^−1^)	Energy Density (J cm^−3^)	Efficiency	Lifetime	Cost	Weight
Dielectric capacitor	10^4^–10^7^	2–30	High	High	Low	Low
Supercapacitor	>10^4^	>20	Low	High	High	High
Lithium-ion batteries	<10^3^	>200	High	Low	Low	Low

**Table 2 membranes-12-00274-t002:** Energy storage properties of the multilayered hierarchical polymer composites.

Structure	Constituent	Breakdown Strength (MV m^−1^)	Energy Density (J cm^−3^)	Efficiency (%)	Refs.
Bi-layered films	P(VDF-TrFE-CFE)/PI	487.5	9.6	58	[31]
Tri-layered films	PVDF/P(VDF-TrFE-CTFE)/PVDF	599	20.86	62	[34]
	P(VDF-HFP)/PMMA/P(VDF-HFP)	440	20.3	84	[35]
	DE + P(VDF-HFP)/PMMA/DE + P(VDF-HFP)	300	11.8	89	[36]
	P(VDF-HFP) + PMMA/DE/P(VDF-HFP) + PMMA	350	15	76.5	[39]
	PMMA/DE/PMMA	347.8	12.45	89	[40]
Multilayered films	PVDF//P(VDF-TrFE-CTFE)	426	18.12	80	[41]
	P(VDF-HFP)//PMMA	735	25.3	84.3	[45]
	PVDF//PMMA	630	20.7	63	[46]

## Data Availability

Not Applicable.

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
