# Peer review of "Research Advances in Hierarchically Structured PVDF-Based All-Organic Composites for High-Energy Density Capacitors"

_membranes, 2022, doi:10.3390/membranes12030274_

Round 1
Reviewer 1 Report
The manuscript reported the development of ferroelectric PVDF-based composites for the applications of high-energy-storage capacitors, which is also known as polymer film capacitors. This manuscript mainly focuses on the hierarchically-structured all-organic composites. The regulating strategies of such types of composites are highlighted from the perspective of the preparation approaches, tailored material choices, layer thicknesses, and interfaces. In the end, the remaining problems and challenges are summarized, and an outlook for future energy storage applications are provided.
I consider the content of this manuscript will definitely meet the reading interests of the readers of the Membranes journal. However, the discussion and explanation should be further improved. Therefore, I suggest giving a minor revision and the authors need to clarify some issues. This could be a comprehensive and meaningful work after revision.
Detailed comments are contained in the PDF file.

Reviewer 2 Report
The research aims to deepen the current knowledge available for the development of energy-storage devices. Specifically, the authors' interest was directed to the development of completely organic composites with a hierarchical structure with improved density and energy efficiency and potentially usable in the production of high energy-density capacitors.
The results obtained so far, collected in graphs mainly with poor resolution and tables, are discussed in a language that must necessarily be improved in order to ensure a safe advancement of the current know-how.
Apart from a number of typos still present in the manuscript, some sentences are long or poorly constructed and therefore unclear even for readers familiar with the aspects covered.
For example, lines 52-55, the sentence starting with “For hierarchical nanocomposites…” is too long and partially twisted and, therefore, it is recommended to rephrase it.
Other phrases to review are:
Lines 110-112, 118-121, 147-149 and 363-365.
For the rest, here are some typos:
line 41: please replace “… polymers, and it is able…” with “… polymers. They are able… ".
Line 51: please correct "is" with "are".
Line 193: please correct "centric layer" with "core layer".
Line 198: the term “D-E loops” is introduced for the first time in the text. In this regard, it is advisable to:
- Standardize the explanation of this code shown in the caption of Figures 1 and 6; and
- insert the same explanation also in the text (line 198).
Line 268: please delete the word "with".
Line 310: please replace the word "binding" with the word "combining".
Line 344: the word “theoretically” is superfluous. Please, cancel it.
Overall, the multidisciplinary nature of the research and the foreseeable positive repercussions of the same in the field of energy storage make the work appreciable and relevant from a scientific point of view. However, MINOR REVISIONS are essential to enhance the impact of the same on the scientific community involved currently.
Reviewer 3 Report
Authors reviewed organic composites for high energy density capacitors. Authors claimed in this review, most research advances of hierarchically structured all-organic composites for the energy storage application are systematically classified and summarized. But I found they have only focused on the few organic compounds like PVDF, PMMA only while missing other important organic compounds such as conducting polymers and their composite. Otherwise, authors could revise the title based on their focus. authors should discuss the energy density and properties of hierarchically structured composite versus simpler structured composite material. I suggest seeing this related literature Journal of Energy Storage 33, 102080. I recommend publishing after minor corrections.
